# Monounsaturated Fatty Acids in Cardiovascular Disease: Intake, Individual Types, and Content in Adipose Tissue as a Biomarker of Endogenous Exposure

**DOI:** 10.3390/nu17152509

**Published:** 2025-07-30

**Authors:** Jonas Pedersen, Berit Storgaard Hedegaard, Erik Berg Schmidt, Christina C. Dahm, Kirsten B. Holven, Kjetil Retterstøl, Philip C. Calder, Christian Bork

**Affiliations:** 1Department of Cardiology, Regional Hospital Viborg, 8800 Viborg, Denmark; jonpe5@rm.dk (J.P.); berit.hedegaard@rm.dk (B.S.H.); 2Department of Clinical Medicine, Aalborg University, 9000 Aalborg, Denmark; 3Department of Public Health, Aarhus University, 9000 Aarhus, Denmark; ccd@ph.au.dk; 4Department of Nutrition, Institute of Basic Medical Sciences, University of Oslo, 0317 Oslo, Norway; k.b.holven@medisin.uio.no (K.B.H.);; 5National Norwegian Network on Familial Hypercholesterolemia, Department of Endocrinology, Morbid Obesity and Preventive Medicine, Oslo University Hospital, 0317 Oslo, Norway; 6The Lipid Clinic, Medical Department, Oslo University Hospital, 0317 Oslo, Norway; 7School of Human Development and Health, Faculty of Medicine, University of Southampton, Southampton SO16 6YD, UK; p.c.calder@soton.ac.uk; 8NIHR Southampton Biomedical Research Centre, University Hospital Southampton NHS Foundation Trust and University of Southampton, Southampton SO16 6YD, UK; 9Department of Cardiology, Aalborg University Hospital, 9000 Aalborg, Denmark

**Keywords:** monounsaturated fatty acids, atherosclerotic cardiovascular disease, diet, dietary patterns, adipose tissue, endogenous exposure of MUFA

## Abstract

Unhealthy dietary patterns are a major modifiable risk factor for atherosclerotic cardiovascular disease (ASCVD). International guidelines recommend reducing saturated fatty acid intake while increasing polyunsaturated and monounsaturated fatty acids (MUFAs) to mitigate cardiovascular risk. However, evidence regarding MUFAs and risk of ASCVD remains conflicting, with recent studies raising concern about a potential higher risk associated with MUFA intake. The aim of this narrative review is to provide an overview of current knowledge and gaps in the literature regarding MUFAs and the risk of ASCVD with a focus on intake, individual types, and content in adipose tissue as a biomarker of endogenous exposure. Main findings reveal that most studies have inappropriately combined all MUFAs together, despite individual MUFA types having different biological effects and showing varying correlations between dietary intake and adipose tissue content. Adipose tissue composition may serve as a biomarker of long-term MUFA exposure, reflecting cumulative intake over one to two years while minimizing biases inherent in dietary assessments. However, tissue levels reflect both dietary intake and endogenous synthesis, complicating interpretation. Importantly, the source of MUFAs appears critical, with plant-derived MUFAs potentially offering advantages over animal-derived sources. In conclusion, we suggest that future research should focus on individual MUFA types rather than treating them as a homogeneous group, investigate their specific dietary sources and associations with ASCVD risk, and use adipose tissue biomarkers to improve exposure assessment and clarify causal relationships while considering overall dietary patterns.

## 1. Introduction

Unhealthy diets may contribute to more than half of the atherosclerotic cardiovascular disease (ASCVD) burden in Europe [1]. Therefore, a healthy diet is the cornerstone for prevention of ASCVD, and international guidelines universally recommend an intake of foods with a healthy composition of fatty acids (FAs): a limited intake of foods high in trans fatty acids and saturated fatty acids (SFAs) with a higher proportion of fat derived from polyunsaturated fatty acids (PUFAs) and monounsaturated fatty acids (MUFAs) [2,3]. However, previous studies have shown conflicting results regarding the role of MUFAs in prevention of ASCVD, and recent studies have raised concern that MUFAs might be associated with a higher risk of ASCVD. These overall findings are based on twenty studies, of which, as of now, only one [4] is included in the current guidelines [2,3,4,5,6,7,8,9,10,11,12,13,14,15,16,17,18,19,20,21,22]. This is of major public health relevance, as 10–20% of the total energy intake in most humans originates from MUFAs [23].

FAs structurally consist of a hydrocarbon chain with a carboxyl group at one end and a methyl group at the other. The chain length, number of double bonds, location of double bonds in the chain, and configuration of the double bond(s)—*cis* or *trans*—differ, with SFAs having carbon atoms linked entirely by single bonds, whereas MUFAs have one double bond and PUFAs have more than one double bond in the carbon chain [24]. These structural characteristics define each individual FA and determine their biological properties and health effects. Upon intake, FAs may become available for energy production, incorporation into cell membranes and pools for storage (e.g., adipose tissue) or further endogenous metabolism, leading to formation of other FAs and biologically active substances [25,26]. The content of MUFAs in plasma, cell membranes or adipose tissue represents biomarkers of MUFA exposure, with limited concern for measurement error, e.g., recall or social desirability bias. The content of MUFAs in plasma or erythrocytes may reflect intake and metabolism of MUFAs over a period of days to weeks [26], while the content of MUFAs in adipose tissue probably reflects intake and metabolism of MUFAs during the preceding 1 to 2 years [26], and is therefore the preferred biomarker in studies of ASCVD and other chronic diseases. Furthermore, adipose tissue is an active organ that may release FAs for utilization in the body, and the content of FAs in adipose tissue may therefore also be considered a biomarker of the endogenous exposure of individual FAs [27,28,29].

The aim of the paper is to provide a review of the role of MUFAs in ASCVD with focus on individual types of MUFAs, different dietary sources of intake, and content in adipose tissue, as a biomarker of endogenous exposure in humans.

## 2. Dietary Sources of MUFAs

There are several types of MUFAs with different carbon chain length, double bond configuration, and food sources (Table 1). Oleic acid constitutes 90% of total MUFA intake in most Western populations, and foods rich in oleic acid include vegetables oils such as olive oil and rapeseed oil, as well as many foods of both plant and animal origin. In contrast, foods rich in other MUFAs like vaccenic acid include ruminant meat and dairy products, whereas major sources of palmitoleic acid are fatty fish, nuts and sea buckhorn. Erucic acid is found in rapeseed oil, fish and jojoba oil, whereas cetoleic acid is found in herring and jojoba oil. Limited data exist on dietary sources of myristoleic acid. Gadoleic acid is found in cod liver oil and jojoba oil, whereas gondoic acid is reported in cod liver oil, but more comprehensive studies of different food sources of most MUFAs are needed (Table 1).

A diet known to be high in MUFAs is the Mediterranean diet, which has been associated with cardioprotective effects and a lower risk of ASCVD [14]. A major source of MUFAs in the Mediterranean diet is olive oil [30]. However, other major components of the traditional Mediterranean diet include non-starchy legumes, cereals, fruits, vegetables, nuts, fish, and wine, making it difficult to know which specific part of the diet may be responsible for the beneficial health effects [31]. The PREDIMED study investigated the effect of adhering to a Mediterranean diet supplemented with either extra virgin olive oil or nuts, compared to a diet reduced in fat [14,24]. The study found a significantly lower incidence of ASCVD in the groups assigned to the Mediterranean diet [14]. Interestingly, individuals assigned to the Mediterranean diet had an intake of total MUFA markedly higher than those assigned to a low-fat diet.
nutrients-17-02509-t001_Table 1Table 1Types of MUFAs and relevant food sources.NameKnown Dietary SourceOleic acid (18:1ω-9)Vegetable oils (e.g., olive oil, rapeseed oil, sunflower oil), many plant and animal foods [22,24,32,33].Vaccenic acid (18:1ω-7)Ruminant meat and dairy products [34] *, cod liver oil [33]Palmitoleic acid (16:1ω-7)Fatty fish (e.g., salmon) and fish oils (e.g., cod liver oil), nuts (e.g., macadamia nuts and their oil), sea buckhorn [33,35].Myristoleic acid (14:1ω-5)Gap in literature. No comprehensive studies on food sources.Gadoleic acid (20:1ω-9)Cod liver oil and jojoba oil [33,36]. Otherwise, gap in literature. No comprehensive studies on food sources.Gondoic acid (20:1ω-11)Cod liver oil [33]. Otherwise, gap in literature. No comprehensive studies on food sources.Erucic acid (22:1ω-9)Rapeseed oil, jojoba oil and fish [33,36,37,38,39].Cetoleic acid (22:1ω-11)Herring oil, cod liver oil, and jojoba oil [33,36,40].***** Mainly as a *trans*-isoform. Abbreviations: MUFAs = monounsaturated fatty acids.

A meta-analysis of 172,891 participants from three large prospective cohorts (the European Prospective Investigation into Cancer and Nutrition-CVD study, the UK Biobank study, and the INTERVAL study) showed that the main sources of MUFAs were vegetable oils, pasta, rice and other grains, butter, and cheese [41]. However, regional differences may further affect total MUFA intake. In Southern Europe, vegetables oils are the main dietary source of MUFAs, whereas in Northern Europe, MUFA are often derived from meat and dairy products [42].

Information on major food sources of MUFAs, as well as the individual intake and their content in adipose tissue, was collected in 3559 middle-aged Danish men and women in the Diet, Cancer and Health cohort [43,44]. Information on diet was collected between 1993 and 1997 using a validated food frequency questionnaire [44], and a gluteal adipose tissue biopsy sample was collected in all participants, and later analyzed by gas chromatography [45]. In this cohort, the median intake of MUFAs was 27.5 g/day. The major sources of MUFAs included red meat, margarines and processed meat (Table 2). Oleic acid was the most abundant dietary MUFA (median intake 23.0 g/d), followed by vaccenic acid (1.4 g/d), palmitoleic acid (1.4 g/d) and myristoleic acid (0.4 g/d) (Table 3).

The major food sources of MUFAs (and other macronutrients) in a given population serve as underlying dietary patterns. These patterns can be evaluated using radar plots by visualizing the intake of selected foods, stratified by levels of MUFA intake [46]. The underlying dietary pattern of MUFA intake in the Diet, Cancer and Health cohort is shown in Figure 1A, which may be used to describe MUFA as an indicator of the underlying diet and to evaluate confounding data from the diet [43].

Participants in the highest quintile of MUFA intake had a higher intake of fish, red meat, processed meat, other animal fat, eggs, margarines and mayonnaises, and a lower intake of lean dairy products and fruits and vegetables, compared to participants in the lowest quintile of MUFA intake.

## 3. Endogenous Pathway of MUFA Synthesis and Metabolism

Blood and tissue MUFAs may originate from the diet or from endogenous synthesis from carbohydrates and SFA. A simplified overview of the pathway of synthesis and metabolism of MUFAs is shown in Figure 2.

The first key enzyme in the MUFA biosynthesis is the stearoyl-CoA desaturase 1 (SCD1), also known as Δ9D [47]. SCD1 catalyzes MUFA production by introduction of a double bond to the SFA palmitic acid (16:0) and stearic acid (18:0) to generate palmitoleic acid (16:1ω-7) and oleic acid (18:1ω-9), respectively. Thus, SCD1 activity may be of major importance for the endogenous synthesis of MUFAs and previous findings have indicated that SCD1 activity may influence ASCVD risk [48]. SCD1 activity depends on dietary factors, as a high carbohydrate consumption upregulates SCD1 expression via insulin signaling, while replacing dietary SFAs with MUFAs seems to suppress SCD1 activity, and adipose tissue SCD1 expression may be further downregulated in subjects with obesity [35,49,50,51]. In addition, genetic factors affecting SCD1 activity may also be important [48].

Palmitoleic acid (16:1ω-7) may further be elongated to vaccenic acid (18:1ω-7) catalyzed by the fatty acid elongase 5 (ELOVL5). In mammalian cells, ELOVL5 activity seems to depend on SCD1-driven availability of palmitoleic acid [52].

Oleic acid (18:1ω-9) may be elongated to gadoleic acid (20:1ω-9) and further to erucic acid (22:1ω-9) or the isoform cetoleic acid (22:1ω-11) and nervonic acid (24:1ω-9), through enzymatic elongations [53,54,55]. There is no compelling evidence that human cells are able to endogenously synthesize cetoleic acid in appreciable amounts, but exogenously supplied cetoleic acid improves the efficiency of the *n*-3 PUFA metabolic pathway [56,57]. Furthermore, oleic acid may be desaturated into a specific type of linoleic acid (18:2ω-9) by the Δ-6-desaturase (Δ6D) and, through further desaturation, may be converted to mead acid (20:3ω-9) down a pathway of ω-9 series of PUFA [58].

The complex endogenous biosynthesis of MUFA complicates the interpretation of studies considering total MUFA intake as an exposure, and calls for the investigation of the different individual MUFAs based on estimated intakes and tissue levels.

## 4. The Content of MUFAs in Adipose Tissue and Correlations with MUFA Intake

Limited data exist regarding intake of individual MUFAs and their subsequent content in adipose tissue. Few studies have reported the content of individual MUFAs in adipose tissue. Oleic acid (~44% of total FA) was the most abundant MUFA in adipose tissue biopsies in the Diet, Cancer and Health cohort, followed by palmitoleic acid (~7% of total FA), vaccenic acid (~2% of total FA), gadoleic plus gondoic acid (~1% of total FA), myristoleic acid (~0.4% of total FA), and erucic acid (0.05% of total FA) (Table 3).

Human adipose tissue serves as a dynamic reservoir for dietary FAs, and thus may represent overall FA exposure in the body through a combination of dietary intake and endogenous production [26,28,29,59]. Correlations between intake of MUFAs obtained from food frequency questionnaires (FFQs) and adipose tissue content of MUFAs are generally weak to modest, but may differ for individual MUFAs [28,60,61]. Thus, the MUFA content of adipose tissue biopsies from 188 individuals correlated to MUFA intake assessed by two 7 d weighed-diet records, with Pearson correlations of 0.26 for women and 0.04 for men [28]. Another study investigated the content of MUFAs in adipose tissue and MUFA intake (as energy percentage) assessed via a 170-item FFQ in 35 individuals, and found a Spearman rank correlation coefficient of 0.19 for total MUFA and 0.38 for oleic acid [61]. Palmitoleic acid in adipose tissue was inversely correlated (Spearman rank correlation: −0.14) with intake of palmitoleic acid expressed in energy percentage, but the specific isoform of palmitoleic acid was not described [61]. A study of 115 women presented statistically non-significant correlations between MUFA intake and content in adipose tissue [60]. Such correlations may also depend on limitations of the FFQ itself, the food composition database used, and the tissue sampling sites, as analyses of perirenal, abdominal and gluteal adipose tissue from 143 autopsies revealed that subcutaneous fat contained higher proportions of oleic acid and palmitoleic acid than perirenal and abdominal adipose tissue [62]. These findings are consistent with other analyses of MUFA composition in visceral adipose tissue [63,64], yet different sample sites of visceral adipose (atrial vs. epicardial) tissue have shown variations in the composition of MUFAs [65]. Though the composition of FA visceral (epicardial) adipose tissue may contribute to risk of ASCVD [66], there is a gap in the literature regarding visceral adipose tissue content of MUFAs and the risk of ASCVD. However, analyses of FA composition of adipose tissue in dietary studies usually investigate subcutaneous biopsies taken from the gluteal region. Meanwhile, a study of 4114 men and women found that women had higher proportions of MUFAs in adipose tissue, compared to men—who had higher proportions of SFAs [67]. In rodents, no sex differences have been observed in the endogenous synthesis of MUFAs in adipose tissue [68]. However, differences in the metabolism of MUFAs in humans and its potential link to ASCVD remains as a gap in the literature.

Correlations in the Diet, Cancer, and Health cohort between the MUFA content in adipose tissue and dietary intake of MUFAs expressed in absolute intake (g/d) and as a percentage of total FA intake are shown in Table 4. The correlations between the absolute intakes of MUFAs and the content in adipose tissue were rather weak. However, positive correlations were found when intake of MUFAs was expressed as a percentage of total FAs. Thus, high correlations were found between intake of myristoleic acid (0.43) and gadoleic/gondoic acid (0.34), whereas modest correlations were found for oleic acid (0.22) and palmitoleic acid (0.14). No appreciable correlations were found for erucic acid (0.10) and vaccenic acid (−0.06).

Doubling dietary intake of MUFAs from olive oil while reducing intake of SFAs by half in 20 participants in a controlled feeding trial showed a 0.70% absolute increase in adipose tissue oleic acid content and an 0.45% absolute increase in total MUFA content after eight weeks of dietary intervention [29]. This was a subgroup analysis, and the tissue sampling site was periumbilical [29]. In the short-term, diet-derived oleic acid incorporates into adipose tissue within hours [29,69,70,71].

The underlying dietary pattern according to adipose tissue content of MUFAs in the Diet, Cancer and Health cohort is shown in Figure 1B [43]. The differences in the underlying dietary pattern were less pronounced for intake of selected foods in the highest and lowest quintile of MUFAs in adipose tissue than for intake of MUFAs, which might indicate that confounding data from the diet may be of lesser concern for adipose tissue content of MUFAs than for intake of MUFAs. However, participants in the highest quintile of adipose tissue content of MUFA had a higher intake of animal fat and a lower intake of cereals, lean dairy products, margarines and vegetable oils, compared to participants in the lowest quintile of MUFAs in adipose tissue.

The varying correlations across studies may reflect both methodological and biological variability. Thus, the adipose tissue content of MUFAs reflects an objective biomarker of MUFA exposure which may not be equivalent to those reported in FFQs, due to recall or social desirability bias. Also, correlations may differentiate between predominantly diet-derived MUFAs and those with substantial endogenous synthesis. Importantly, weak correlations do not necessarily imply a lack of biological relevance; rather they may highlight the complexity of FA intake, metabolism, and storage.

## 5. MUFAs and the Risk of ASCVD

Atherosclerosis is the main underlying disease process that may result in development of ASCVD and clinical symptoms. The atherosclerotic process is extremely complicated, but central components include endothelial dysfunction, retention of (oxidized or otherwise modified) low-density-lipoprotein (LDL)-cholesterol in the sub-endothelial layer and inflammatory responses [72,73]. MUFAs may influence several pathways involved in the pathogenesis of ASCVD, although the detailed effects of MUFAs are not fully understood. However, some studies have suggested that MUFAs may modulate the atherosclerotic process through preservation of hepatic LDL-receptors and cholesterol clearance [74,75], regulation of hepatic cholesterol synthesis [76,77], lowering of LDL-cholesterol and anti-inflammatory effects [57,78].

The majority of research examining the effects of MUFAs on atherosclerosis and ASCVD has predominantly targeted oleic acid, leaving the biological activities of other MUFA types comparatively underexplored. Substituting SFAs with oleic acid has thus been documented to reduce total and LDL-cholesterol plasma concentrations [79,80]. Oleic acid also seems to render LDL more resistant to oxidation, thereby mitigating the formation of proatherogenic oxidized LDL-cholesterol [81]. Furthermore, when compared to PUFA, oleic acid exerts a decreased peroxidizability in lipoproteins and cell membranes, which, given the proinflammatory nature of oxidative stress, would be expected to attenuate inflammatory responses [30]. However, while oleic acid is often characterized as having “anti-inflammatory” effects, its impact on inflammation may be limited overall [82]. Additional evidence has suggested that oleic acid exerts a modest antihypertensive effect [83] and may improve glycemic regulation and insulin sensitivity [84,85]. Overall, dietary supplemented oleic acid may exert multiple properties that together mitigate atherogenesis [86]. Olive oil, which—depending on olive cultivars—consists of more than 70% oleic acid, has been associated with a spectrum of cardioprotective outcomes, including a lower risk of type 2 diabetes and diminished inflammation; however, other constituents of olive oil such as polyphenols and phytosterols may also play significant roles in these health benefits [87,88].

Other MUFAs have been far less investigated than oleic acid, but studies have suggested that palmitoleic acid may have a role in improving insulin sensitivity, lowering hepatic lipid accumulation, and attenuating atherogenesis [89,90]. Furthermore, inflammation may be attenuated by palmitoleic acid via downregulation of the expression of the proinflammatory transcription factor NFκB and the production of proinflammatory mediators such as interleukin-1β [91,92,93,94]. Supplementation with palmitoleic acid has also been linked to decreases in total cholesterol, LDL-cholesterol, and triacylglycerol levels [95,96]. In addition, studies have shown that circulating levels of palmitoleic acid may be associated with improved insulin sensitivity and a lower risk of type 2 diabetes [97,98]. These effects of palmitoleic acid in humans are under further investigation [99].

Cis-vaccenic acid accumulated in red blood cell membranes has been inversely associated with atherosclerosis [100]. Yet, the biological mechanisms of vaccenic acid seem mostly described for its *trans*-isomer: dysregulated postprandial lipid metabolism contributes to ASCVD risk, and *trans*-vaccenic acid has demonstrated the capacity to lower postprandial lipids, triacylglycerols, and chylomicrons, suggesting a potential attenuation of atherogenesis [101]. However, a major clinical trial [102] indicated that *trans*-vaccenic acid exerts distinct unhealthy effects on lipoprotein metabolism compared to industrial trans fats, notably increasing total and LDL-cholesterol, as well as high-density-lipoprotein cholesterol and lipoprotein(a). *Trans*-vaccenic acid may also exert anti-inflammatory properties by suppressing vascular cellular adhesion molecule-1 and intra-cellular adhesion molecule-1, both key pro-atherogenic and proinflammatory adhesion molecules [103,104]. Future clear distinction between *cis*- and *trans*-isomers of vaccenic acid in studies is needed to avoid misclassification of biological properties.

The biological role of myristoleic acid in atherogenesis remains insufficiently characterized. It has been suggested that dietary myristoleic acid is inversely associated with lipoprotein(a), a causal ASCVD risk factor, but this needs confirmation [105,106,107].

Long-chain MUFAs (LCMUFAs) including gadoleic, gondoic, erucic, and cetoleic acid have been shown to reduce systemic inflammation, likely via beneficial alterations in proteins involved in complement activation, blood coagulation, and lipid metabolism [108]. LCMUFAs have also been reported to lower plasma LDL-cholesterol, very-low-density-lipoprotein and triglycerides, as well as improving insulin resistance [109,110,111]. Finally, a meta-analysis of cetoleic acid supplementation from fish oil in rodent models supports a cholesterol-lowering effect [112].

Thus, there are several mechanisms by which various MUFAs could potentially influence atherosclerosis and the development of ASCVD, but the biological actions of individual MUFAs still warrant further investigation.

MUFAs have been far less studied than SFAs and PUFAs in relation to risk of ASCVD [2,7]. Existing studies investigating the role of MUFAs in the risk of ASCVD have shown conflicting results, with neutral associations [5,6,8,9,10,11,12,22], inverse associations [4,13,14] and recent studies raising concern that MUFAs might be associated with an increased risk of ASCVD [15,16,17,18,19,20]. A limitation of the majority of these studies is the reliance on estimated MUFA exposure obtained through dietary intakes collected using FFQs. These are prone to measurement error, due to potential response bias, recall bias and social desirability bias, as well as limitations in the food composition databases used to calculate MUFA content of reported foods. The majority of previous cohort studies investigating the association between MUFAs and risk of ASCVD outcomes were established more than 20 years ago, which may be a weakness. Dietary habits, including sources of MUFAs, may have changed during long-term follow-up, which may be a possible explanation for the inconsistent findings observed. Furthermore, the sources of different MUFAs in various populations is likely important, as well.

An FFQ-based prospective cohort of 158,198 individuals, for whom oleic acid contributed more than 80% of total intake of MUFAs, with a median follow-up of 3.4 years found no association between total intake of MUFAs and ASCVD risk [22]. Looking further at individual MUFAs, higher intakes of palmitoleic acid and *cis*-vaccenic acid were associated with a higher risk of ASCVD [22]. In contrast, a modelled substitution analysis of a cohort of 93,384 individuals showed a lower risk of ASCVD by replacing SFAs with plant-derived MUFAs, rather than animal-derived MUFAs [113], indicating the importance of considering individual MUFAs and their respective food sources. Given that MUFAs are ubiquitous in any foods containing fat, the confounding data from food sources of MUFAs (plant vs. animal) is likely a significant factor in the conflicting findings from these studies [25].

Studies investigating the associations of circulating MUFAs with ASCVD have also shown conflicting results [17,18,19,20,21]. A prospective cohort of 89,242 individuals found a positive association between plasma levels of MUFA and risk of ASCVD, but results became neutral when adjusting for triglycerides [17]. When integrating genetic risk scores specifically for coronary artery disease, plasma MUFA levels were positively associated with ASCVD in 101,367 individuals, independent of genetic risk, but the sources of MUFAs were mostly animal-based [19]. A meta-analysis of five cohorts (the British Women’s Heart and Health Study (BWHHS) [114], the British Regional Heart Study [115], the Whitehall II Study [116], Southhall and Brent Revisited (SABRE) [117], and the Caerphilly Prospective Study [118]) and one case-control study (the United Kingdom Collaborative Trial of Ovarian Cancer Screening [119]) found circulating MUFAs consistently linked to a higher risk of ASCVD [20]. Similar results were found in another meta-analysis [18] which also included BWHHS [114], SABRE [117], and the FINRISK 1997 study [120]. A Mendelian randomization analysis based on genome-wide association studies on MUFAs found no relationship between genetically predicted circulating MUFAs and ASCVD risk [21]. However, circulating levels of MUFAs may only represent the exposure of MUFAs for a few days to weeks [26,29]. In contrast, analyses of FA composition in adipose tissue may represent long-term exposure of one to three years preceding the biopsy [26,59,121]. In another case-control study (1828 cases and 1828 matched controls), higher adipose tissue content of palmitoleic acid was inversely associated with the risk of non-fatal myocardial infarction [122]. Looking specifically at ischemic stroke, two meta-analyses found that the association between a higher intake of MUFAs and ischemic stroke was neutral, but an inverse association was seen with haemorrhagic stroke [5,6].

## 6. Conclusions and Perspectives

Current guidelines recommend that MUFAs should replace intake of SFAs to lower the risk of ASCVD, but studies investigating the associations between MUFA intake and the risk of ASCVD have shown conflicting results, with recent studies suggesting that MUFA intake may be associated with a higher risk. However, most studies have not differentiated between individual MUFAs, which may be of importance due to differences in biological effects and underlying dietary sources. Based on this narrative review, practical dietary guidelines, in alignment with current guidelines, should consider the food source of MUFAs with an apparent focus on plant-derived MUFAs. To support the current dietary guidelines, further studies are needed. Combining MUFAs together, as was done in most studies previously, may be too simplistic to determine the biological effects of different FAs, which is important for guiding specific dietary recommendations, and further research on the role of individual MUFAs for development of ASCVD is needed. Adipose tissue content represents the long-term exposure of FAs, with the existing literature suggesting that MUFA levels represent intake from the previous one to two years. However, further studies are needed to confirm whether the turnover of MUFAs in adipose tissue follows a similar time pattern to that of other FAs. The correlations between dietary intake of MUFAs and the content of MUFAs in adipose tissue vary for each individual MUFA. In adipose tissue, it is not clear which MUFA may serve as a potential biomarker of the risk of ASCVD, and this is complicated because each individual MUFA possesses different biological effects in the development of ASCVD. Future investigation of tissue levels of MUFAs may, however, provide valuable insight into the associations between individual MUFAs and ASCVD risk. The interpretation of biomarker studies, though, may be complicated, because tissue levels of MUFAs may reflect dietary intake as well as endogenous synthesis. Furthermore, the sources of MUFAs may be of importance, and investigation of MUFAs from different sources indeed warrants further investigation. Finally, when investigating effects of individual FAs and food items, it is of major importance to be aware of, and study, the overall dietary patterns.

This review suggests the following focus points for future research:(1)Investigation of the biological effects of individual types of MUFAs on the development of atherosclerosis. Preferably, this should be carried out in human studies, but animal studies may be of value.(2)Investigation of the importance of food sources for individual MUFAs and their associated risk of ASCVD. Preferably, this should be carried out in clinical trials, but food science studies investigating the MUFA effects and composition of different foods would be valuable, too.(3)Investigation of the associations between total and individual MUFA and ASCVD risk using complementary measures of exposure, such as estimated dietary intakes and content of MUFAs in blood or, preferably, adipose tissue. Future clinical trials or cohort studies in humans may benefit from using adipose tissue biopsies to indicate MUFA exposure, ideally obtaining several biopsies over time to reflect MUFA turnover and changes in dietary patterns.

The gold standard for investigating the effects of diet on cardiovascular health is randomized clinical trials. While the PREDIMED study was a landmark trial [14], its findings need to be confirmed in other populations with different diets than those of Mediterranean countries. However, clinical trials—especially those of long-term duration required to study ASCVD—are very difficult to perform, expensive, and hard to fund. More basic research regarding the effects of different MUFAs should be undertaken and supported by epidemiological data. Trials investigating the effects of MUFAs (and relevant dietary patterns) on risk factors for ASCVD would then be helpful for designing and conducting new major clinical trials.

## Figures and Tables

**Figure 1 nutrients-17-02509-f001:**
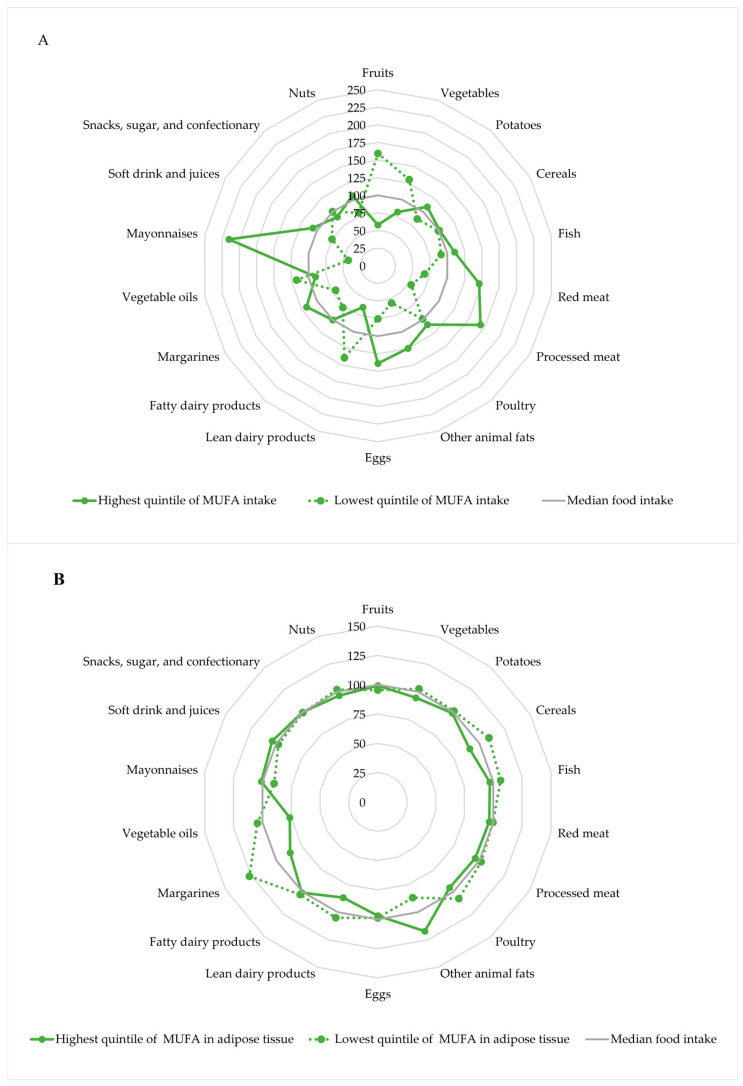
Intake of selected foods according to the intake of MUFA (**A**) and adipose tissue content of MUFA (**B**) in the highest and lowest quintiles, respectively. Intake of foods and MUFA intake was energy-adjusted using the residual method. The radar plots show the underlying dietary pattern according to levels of MUFA indexed to the overall median. The radar plots are based on data from 3559 individuals in the Diet, Cancer, and Health cohort [43].

**Figure 2 nutrients-17-02509-f002:**
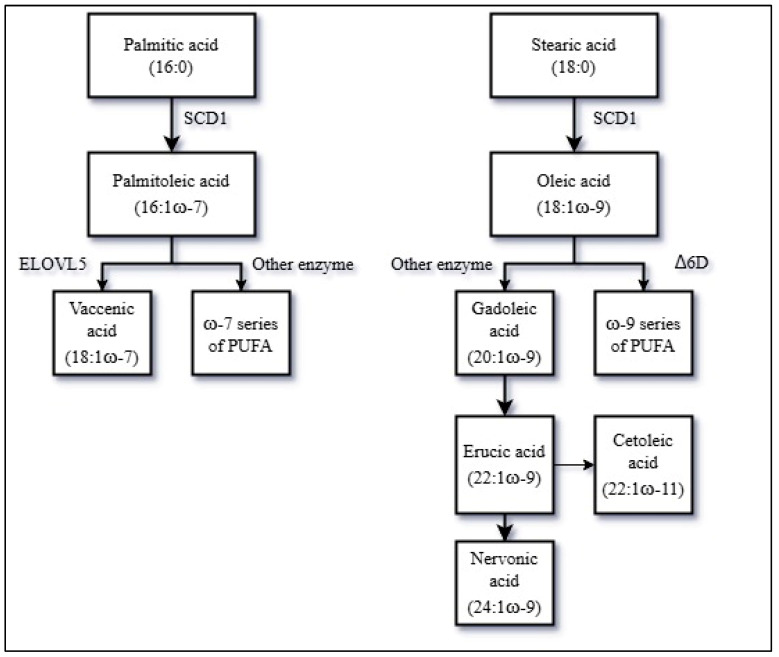
A simplified overview of the endogenous production and metabolism of MUFAs. Abbreviations: MUFAs = Monounsaturated fatty acids, PUFA = Polyunsaturated fatty acids, SCD1 = stearoyl-CoA desaturase 1, ELOVL5 = fatty acid elongase 5, Δ6D = Δ-6-desaturase.

**Table 2 nutrients-17-02509-t002:** Food sources of MUFA in the Diet, Cancer, and Health cohort.

Dietary Source	MUFA Intake in g/d
Fatty dairy products	5.3 (1.2, 14.1)
Red Meat	4.8 (1.9, 10.2)
Margarines	2.8 (0.3, 9.1)
Processed meat	2.4 (0.5, 7.8)
Lean dairy products	1.2 (0.2, 3.2)
Eggs	1.1 (0.3, 3.3)
Vegetable oils	1.1 (0.5, 7.1)
Fish	1.0 (0.2, 3.3)
Mayonnaise and similar products	0.8 (0.1, 6.4)
Snacks	0.7 (0.1, 3.7)
Poultry	0.6 (0.1, 1.7)
Nuts	0.2 (0.0, 1.0)
Vegetables	0.1 (0.0, 0.5)
Fruit	0.1 (0.0, 0.2)
Other animal fat	0.1 (0.0, 0.4)

Presented as medians (5th, 95th percentiles). Data based on 3559 individuals from the Diet, Cancer, and Health cohort [43]. Abbreviations: MUFAs = monounsaturated fatty acids.

**Table 3 nutrients-17-02509-t003:** Dietary intake and adipose tissue content of MUFA in the Diet, Cancer, and Health cohort.

MUFA	Daily Intake of MUFA (% of Total Fat Intake)	Adipose Tissue Content of MUFA (% of Total Fatty Acids)
Total MUFA	27.46 (15.02, 49.28)	54.22 (48.84, 59.35)
Oleic acid (18:1ω-9)	23.02 (12.30, 42.49)	44.02 (40.67, 47.15)
Vaccenic acid (18:1ω-7)	1.41 (0.19, 5.93)	1.99 (1.54, 2.77)
Palmitoleic acid (16:1ω-7)	1.39 (0.76, 2.47)	6.55 (4.15, 9.49)
Myristoleic acid (14:1ω-5)	0.37 (0.16, 0.71)	0.42 (0.24, 0.66)
Gadoleic (20:1ω-9) and Gondoic acid (20:1ω-11)	0.72 (0.34, 1.37)	0.99 (0.72, 1.41)
Erucic acid (22:1ω-9)	0.01 (0.00, 0.05)	0.05 (0.02, 0.10)

Presented as medians (5th, 95th percentiles). Data based on 3559 individuals from the Diet, Cancer, and Health cohort [43]. Abbreviations: MUFAs = monounsaturated fatty acids.

**Table 4 nutrients-17-02509-t004:** Pearsons correlations between MUFA content in adipose tissue and absolute and relative intakes of MUFAs.

	Absolute Dietary Intake of MUFAs	Dietary Intake of MUFAs as Proportion of Total Dietary FA Intake
Total MUFAs	−0.08	0.16
Oleic acid (18:1ω-9)	0.04	0.22
Vaccenic acid (18:1ω-7)	−0.09	−0.06
Palmitoleic acid (16:1ω-7)	−0.10	0.14
Myristoleic acid (14:1ω-5)	0.14	0.43
Gadoleic (20:1ω-9) and Gondoic acid (20:1ω-11)	0.26	0.34
Erucic acid (22:1ω-9)	0.10	0.10

Data based on 3559 individuals from the Diet, Cancer, and Health cohort [43]. Abbreviations: MUFAs = monounsaturated fatty acids, FAs = fatty acids.

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
