# Peer review of "Monounsaturated Fatty Acids in Cardiovascular Disease: Intake, Individual Types, and Content in Adipose Tissue as a Biomarker of Endogenous Exposure"

_nutrients, 2025, doi:10.3390/nu17152509_

Round 1

Reviewer 1 Report

Comments and Suggestions for Authors

The manuscript entitled “Monounsaturated fatty acids in cardiovascular disease: intake, individual types, and content in adipose tissue as a biomarker of endogenous exposure” presents a review of current knowledge on the role of monounsaturated fatty acid intake, dietary sources, metabolism, as well as content in adipose tissue in the development of atherosclerotic cardiovascular disease (ASCVD). The issue described in the present review is important since the Authors address the influence of MUFA on the development of cardiovascular diseases, which are the leading cause of death worldwide. Moreover, the Authors indicate a knowledge gap in this area, along with the enormous impact of the consumed diet on the development of the above disorder.

In my opinion, the review is interesting, but there are several issues that should be clarified/addressed.

  1. The Authors should describe in more detail the connection and/or mechanisms of action between MUFA and ASCVD development, even though current knowledge is mainly based on animal studies. The description (lines: 222 - 226) is very limited and does not provide a broader view to explain the causal relationship between MUFA content and ASCVD development or progression, e.g., the anti-inflammatory effect of MUFA.

  1. The review lacks a description of the pathomechanism/etiology of ASCVD. I suggest that it should be a separate paragraph, which would provide the reader with a deeper understanding of the issue and its possible connections with MUFA.

  1. Furthermore, in the Conclusion, it is not sufficiently clear which of the examined fatty acids in plasma or adipose tissue can serve as a potential biomarker of the risk of ASCVD developing or predispose to such a role. In light of ASCVD prevention or diagnosis, it would be valuable information. If human studies are limited in this respect, maybe animal models would provide some data?

  1. Some of the cohort studies discussed in the review began in the late 1970s, 1980s, or 1990s, which I think warrants comment, as lifestyle and eating habits may have changed significantly compared to today. Does this potentially influence recommendations?

  1. The Authors discuss human studies where the majority of MUFA determinations were conducted on subcutaneous adipose tissue. However, MUFA content in visceral adipose tissue and its connection with ASCVD should also be discussed.  

  1. Are there any significant gender differences concerning MUFA content in adipose tissue, which in turn affect ASCVD development? If yes, please comment on this, since some cohort studies discussed in the manuscript involved both men and women.

  1. The radar plots shown in Figure 1 (A and B) are poorly legible. Their resolution should be improved.

Reviewer 2 Report

Comments and Suggestions for Authors

Journal:

Nutrients (ISSN 2072-6643)

Manuscript ID

nutrients-3730664

Type

Review

Title

Monounsaturated fatty acids in cardiovascular disease: Intake, individual types, and content in adipose tissue as a biomarker of endogenous exposure

Section

Nutritional Epidemiology

Special Issue

Diet, Nutrition and Cardiovascular Health—2nd Edition

___________________________________

OVERALL COMMENTS

The manuscript:

The authors of this manuscript say that unhealthy habits are a critical modifiable risk factor in the prevention of cardiovascular disease, and international guidelines recommend reducing the intake of saturated fatty acids and increasing the intake of polyunsaturated and monounsaturated fatty acids to reduce risks. Based on this, they intended to review current knowledge on the role of monounsaturated fatty acids in atherosclerotic cardiovascular disease with emphasis on dietary sources, monounsaturated fatty acid subtypes, endogenous synthesis, and adipose tissue as a biomarker of endogenous exposure.

TITLE

          The title is adequate.

ABSTRACT

The Abstract lacks a clear objective about the intent of the review. The authors also provide a comprehensive background on the topic and conclude the abstract by stating, "The review will focus on important areas of future research." I suggest including the main findings of this review in the Abstract. In short, I suggest reformulating this section.

_______

KEYWORDS

The authors presented the following keywords:

 Monounsaturated fatty acids; atherosclerotic cardiovascular disease; diets; dietary sources; dietary patterns; adipose tissue; endogenous exposure of MUFA

I suggest:  Monounsaturated fatty acids; atherosclerotic cardiovascular disease; diet; adipose tissue; endogenous exposure of MUFA

INTRODUCTION

In this section, lines 39-41, we can read that “However, previous studies have shown conflicting results regarding the role of MUFA in prevention of ASCVD and recent studies have raised concern that MUFA might be asso-50 ciated with a higher risk of ASCVD [2, 4-20].” Here the authros cite references 4 to 20 to explain a sentence. In the entire text they present 77 references.

First, I believe that there should be more references throughout the entire text. I saw only a few published in 2025. Please consider visiting PubMed, Embase, or Cochrane to find additional references.

Second, I do not see the need to include sixteen references for a sentence with only three lines.

As shown in lines 54-60, the "MUFAs" category includes monounsaturated variants such as oleic acid, vaccenic acid, and elaidic acid, which are found in the 18-carbon fatty acid category, as well as erucic acid, which is also present in the carbon category. Due to the position of the double bond, as well as the position (cis and trans), they can have very different effects in metabolic and physiological terms. As this review suggests, I suggest further investigating this category of fatty acids and exploring their biological effects.

In lines 68-71 we can read that “Furthermore, adipose tissue is an active organ that may release FA for 67 utilisation in the body, and the content of FA in adipose tissue may therefore also be con-68 sidered a biomarker of the endogenous exposure of individual FA.” Please include references in this sentence.

_______

METHODS

Not presented. It is ok since it is a narrative review.

RESULTS and DISCUSSION

          In lines 116-121 we can read that “The major food sources of MUFA (and other macronutrients) in a given population 116 serve as underlying dietary patterns. These patterns can be evaluated using radar plots by visualizing the intake of selected foods, stratified by levels of MUFA intake. The underlying dietary pattern of MUFA intake in the Diet, Cancer and Health cohort is shown in Figure 1A, which may be used to describe MUFA as an indicator of the underlying diet and to evaluate confounding from diet.”

Please include references here. Should it be reference 29? 30?        

Are radar plots (as found in Figure 1) suitable for comparing quintiles?

          The legend of Figure 1 is: “Figure 1. Intake of selected foods according to the intake of MUFA (A) and adipose tissue content 123 of (B) MUFA in the highest at lowest quintile, respectively. Intake of foods and MUFA intake was 124 energy-adjusted using the residual method. The radar plots showed the underlying dietary pattern 125 according to levels of MUFA indexed to the overall median intake. The radar plots are based on data 126 on 3,559 individuals in the Diet, Cancer, and Health cohort. (29).”

Should it be: “Figure 1. Intake of selected foods according to the intake of MUFA (A) and adipose tissue content 123 (B) MUFA in the highest at lowest quintiles, respectively. Intake of foods and MUFA intake was 124 energy-adjusted using the residual method. The radar plots showed the underlying dietary pattern 125 according to levels of MUFA indexed to the overall median intake. The radar plots are based on data from 126 on 3,559 individuals in the Diet, Cancer, and Health cohort [29]?”

          Table 1: Would it be necessary to include elaidic acid? It is a trans acid, but how often is it present in foods?

          It is necessary to improve the quality of Figure 3.

__________

CONCLUSION AND PERSPECTIVES

This section is adequate, but I suggest including:

How can this review contribute to further research?

____________

REFERENCES

          As mentioned before, please include more references.

Reviewer 3 Report

Comments and Suggestions for Authors

The manuscript is scientifically relevant and offers a well-organized review of monounsaturated fatty acids (MUFA) in the context of cardiovascular disease (ASCVD). However, major revisions are needed to improve the clarity, depth, and utility of the review. Key areas requiring attention include:
-Clarifying the biological mechanisms differentiating individual MUFA effects
-Providing a transparent summary of how studies were selected and evaluated
-Addressing confounding from food sources (plant vs. animal MUFA)
-Interpreting the correlation data between dietary intake and tissue content with more nuance
-Strengthening the conclusion with more practical dietary guidance
Once these issues are addressed, the manuscript will be much stronger and well-positioned for publication.

Round 2

Reviewer 1 Report

Comments and Suggestions for Authors

I accept all corrections submitted by the Authors.

Author Response

Thank you for your time.

Reviewer 3 Report

Comments and Suggestions for Authors

This review article explores the relationship between monounsaturated fatty acids (MUFAs) and atherosclerotic cardiovascular disease (ASCVD), an important and timely topic in preventive nutrition. The manuscript effectively highlights the conflicting evidence surrounding MUFA intake and cardiovascular risk, while emphasizing the public health significance given MUFAs’ high contribution to energy intake in many populations.

The description of MUFA metabolism, endogenous synthesis, and the role of adipose tissue as a biomarker is particularly strong. The authors also successfully contrast short-term biomarkers (e.g., plasma levels) with long-term markers such as adipose tissue, which provides more stable measures of dietary exposure.

However, some improvements in scientific phrasing and sentence structure are needed. Certain terms (e.g., “systematic availability”) should be corrected (“systemic availability”), and grammatical refinements would enhance clarity. Additionally, the discussion on future research directions would benefit from being more specific.

With these minor revisions, the manuscript would be a valuable addition to the literature on dietary fats and cardiovascular health.
